# UPLC-QTOF/MS-Based Nontargeted Metabolomic Analysis of Mountain- and Garden-Cultivated Ginseng of Different Ages in Northeast China

**DOI:** 10.3390/molecules24010033

**Published:** 2018-12-21

**Authors:** Hailin Zhu, Hongqiang Lin, Jing Tan, Cuizhu Wang, Han Wang, Fulin Wu, Qinghai Dong, Yunhe Liu, Pingya Li, Jinping Liu

**Affiliations:** Research Center of Natural Drugs, School of Pharmaceutical Sciences, Jilin University, Fujin Road 1266, Changchun 130021, China; 13578965875@163.com (H.Z.); linhq17@mails.jlu.edu.cn (H.L.); tanjing17@mails.jlu.edu.cn (J.T.); wangcz15@mails.jlu.edu.cn (C.W.); hanw17@mails.jlu.edu.cn (H.W.); wufl17@mails.jlu.edu.cn (F.W.); dongqh17@mails.jlu.edu.cn (Q.D.); lyh133700@163.com (Y.L.)

**Keywords:** mountain-cultivated ginseng, identification, metabolomic analysis, UPLC-QTOF-MS

## Abstract

Aiming at further systematically comparing the similarities and differences of the chemical components in ginseng of different ages, especially comparing the younger or the older and mountain-cultivated ginseng (MCG), 4, 5, 6-year-old cultivated ginseng (CG) and 12, 20-year-old MCG were chosen as the analytical samples in the present study. The combination of UPLC-QTOF-MS^E^, UNIFI platform and multivariate statistical analysis were developed to profile CGs and MCGs. By the screening analysis based on UNIFI, 126 chemical components with various structural types were characterized or tentatively identified from all the CG and MCG samples for the first time. The results showed that all the CG and MCG samples had the similar chemical composition, but there were significant differences in the contents of markers. By the metabolomic analysis based on multivariate statistical analysis, it was shown that CG_4–6 years_, MCG_12 years_ and MCG_20 years_ samples were obviously divided into three different groups, and a total of 17 potential age-dependent markers enabling differentiation among the three groups of samples were discovered. For differentiation from other two kinds of samples, there were four robust makers such as α-linolenic acid, 9-octadecenoic acid, linoleic acid and panaxydol for CG_4–6 years_, five robust makers including ginsenoside Re_1_, -Re_2_, -Rs_1_, malonylginsenoside Rb_2_ and isomer of malonylginsenoside Rb_1_ for MCG_20 years_, and two robust makers, 24-hydroxyoleanolic acid and palmitoleic acid, for MCG_12 years_ were discovered, respectively. The proposed approach could be applied to directly distinguish MCG root ages, which is an important criterion for evaluating the quality of MCG. The results will provide the data for the further study on the chemical constituents of MCG.

## 1. Introduction

Ginseng, the king of herbs in the Orient, has always received a lot of attention, not only as a therapeutic medicinal herb, but also as a health supplement. According to the different growing environments and diverse cultivation methods, there two kinds of ginseng are distinguished in the Chinese Pharmacopoeia: cultivated ginseng (CG) and mountain-cultivated ginseng (MCG). CG is cultivated artificially in gardens, while MCG is grown for at least 10 years [1,2]. MCG, also called “Lin-Xia-Shan-Shen”, can be regarded as a replacement of wild ginseng. MCG is of better quality than CG and offers more production than wild ginseng [3]. Actually, the adulteration or falsification of the cultivation age of MCG has always been a serious problem in the MCG commercial market. As we all know, the chemical components and biological activities of ginseng with different cultivation ages are distinct [4,5], and more aged ginseng is usually of higher economic value. In an investigation of the characteristic components for distinguishing CG (4–7-year of age) and MCG (with 15-years of growth), 12 compounds, including ginsenoside Ra_3_/isomer, gypenoside XVII, quinquenoside R_1_, ginsenoside Ra_7_, notoginsenoside Fe, ginsenoside Ra_2_, ginsenoside Rs_6_/Rs_7_, malonyl ginsenoside Rc, malonyl ginsenoside Rb_1_, malonyl ginsenoside Rb_2_, palmitoleic acid, and ethyl linoleate were regarded as the characteristic chemical markers for the discrimination [6]. Recently, a UPLC/QTOF- MS-based metabolomics approach was applied to the global metabolite profiling of MCG leaf samples aged from 6 to 18 years, and the authors claimed that the approach could also be applied to discriminate MCG root ages indirectly [7]. It is undoubted that the developed method can be used as a standard protocol for discriminating and predicting MCG leaf ages directly, but there might be some inaccuracy and uncertainty when discriminating MCG root ages indirectly. 

In the past decades, some analytical methods focusing on ginsenosides had been used to distinguish MCG from CG, such as thin layer chromatography (TLC), or high performance liquid chromatography (HPLC) [8,9]. However, these technologies require lots of time and energy, and the results cannot provide a comprehensive or accurate discrimination between them. Currently, untargeted metabolomics, combined with multivariate statistical methods such as principal component analysis (PCA) and orthogonal partial least squares discriminant analysis (OPLS-DA), are widely used to profile diverse classes of metabolites and to better understand the chemical diversity and the multiple pharmacological effects of ginsenosides or ginseng [10,11]. Given the multi-component property, the combination of LC-MS-based metabolomic profiling with multivariate statistical analysis methods was used as a rapid means of characterization and was increasingly applied for analyzing ginseng from different herbs, cultivation environments/areas, cultivation ages or different parts [12,13]. As an example, for different herbs belonging to the same genus, specific biomarkers including chikusetsusaponin IVa, ginsenoside Rf and ginsenoside Rc were selected and verified for ginseng [14]. In another example of different parts analysis, the metabolic profiles of root, leaf, flower bud, berry and seed of ginseng were investigated [12,15]. In addition, the approach for the discrimination of different red ginseng root parts was reported. As a result, fine roots had the highest protopanaxadiol (PPD)/protopanaxatriol (PPT) ratio, which could clearly distinguish the main roots from the lateral roots and fine roots parts [16]. Such analysis was also applied to make metabolite profiling and age discrimination of 4- and 6-year- old red ginseng [17], or 1–6 years ginseng [18]. 

In addition, UNIFI, the automated data processing software, is an integrated informatics platform that possesses the ability to incorporate scientific library into a streamlined workflow, aiming at identifying chemical components from complex raw data [19]. The combination of UPLC separation, Q/TOF-MS detection and UNIFI platform has been frequently applied in the characterization of chemical constituents of herbs [20,21]. 

Normally, CG is harvested after a 4–6 years cultivation period, and MCG is collected at ages of 10–20 years. To develop a more direct and more efficient discrimination method for the cultivation ages and to explore potential age-dependent markers, we chose 4, 5, 6-year-old CG and 12, 20-year-old MCG as the analytical samples in the present study. UPLC-QTOF-MS^E^, UNIFI platform and multivariate statistical analysis were then used to profile these two kinds of ginseng. The aims were to systematically screen the chemical components and to perform the non-targeted metabolomic analysis, and in turn will lay the foundation for the establishment of CG and MCG quality criteria in the future. In one hand, this study will reveal the structural diversity of secondary metabolites and the different patterns in CG and MCG. In the other hand, the present study could provide a reference point for a reliable, accurate method for distinguishing among CG and MCG samples of different ages.

## 2. Materials and Methods

### 2.1. Materials and Reagents

A total of 40 batches of CG and MCG root products, including 24 batches of CGs and 16 batches of MCGs, were collected from different cultivation areas in Jilin Province, the main source of ginseng in China. A detailed sample list is given in Table 1. All samples were harvested and collected by Professor Li Ping-ya from Jilin University Institute of Frontier Medical Science, according to China Pharmacopoeia (2015 version) [22]. Voucher specimens have been deposited at the Research Center of Nature Drug, School of Pharmaceutical Sciences, Jilin University, Changchun, China. 

Acetonitrile, methanol were all UPLC-MS pure grade (Fisher Scientific Inc., Geel, Belgium). Formic acid (MS grade) was purchased from Sigma-Aldrich (St. Louis, MO, USA). Leucine enkephaline was provided by Waters (Waters Technologies, Milford, MA., USA). Distilled water was prepared in-house via a Millipore water purification system (Millipore, Billerica, MA, USA). All other chemicals were analytical grade. For reference substances, ginsenoside F_1_ (R20151040), -F_2_ (R20151040), notoginsenoside R_1_ (R20170210), notoginseno- side R_4_ (R20170212) were provided by the Research Center of Natural Drugs, School of Pharmaceutical Sciences, Jilin University, China. Ginsenoside Rb_1_, -Rb_2_, -Rb_3_, -Rc, -Rd, -Re, -Rf, -F_5_, -Rg_1_, 20(*R*)-Rg_2_, 20(*S*)-Rh_1_, 20(*R*)-Rh_1_, 20(*S*)-Rg_3_, 20(*R*)-Rg_3_, -Ro, gypenoside XVII, ginsenoside Rs_1_, -Rs_2_ were isolated in our laboratory and identified by spectroscopic data. Adenine (101774299), tryptophane (73-22-2), palmitoleic acid (101491588) were purchased from Sigma-Aldrich. Notoginsenoside Fe (8105-29-5), D-adenosine (110879- 200502), histidine (624-200304) were purchased from the National Institutes for Food and Drug Control. Ginsenoside Rg_5_ (wkq16051002, Victory Biological Technology Co., Ltd., Sichuan, China), α-linoleic acid (B21469; Yuanye Biological Technology Co., Ltd., Shanghai, China), D-arginin (130701; Nuoye Biological Engineering Co., Ltd., Anhui, China) and phenylpropionic acid (A20160211), quillaic acid (A20171109) were purchased from Beijing Zhongke Quality Inspection Biotechnology Co., Ltd. (Beijing, China) with the Chinese National Standard Sieve No. 3 (R40/3 series). 

### 2.2. Sample Preparation and Extraction 

All the CG and MCG samples were air-dried, grinded (Baijie Stainless Steel Grinder, BJ-800A, Deqing Baijie Electric Apllicance Co. Ltd., Zhejiang, China) and sieved (Chinese National Standard Sieve No. 3, R40/3 series) to get the homogeneous powder respectively. Then, the powder of 40 samples (200 mg accurately weighed per sample) were refluxed respectively with 85% methanol (2 L) at 80 °C for three times (2 h, 2 h, 1 h each time, respectively). Then, the extracts of each sample were combined, concentrated and evaporated to dryness. Each powder was dissolved in 5.0 mL of 80% methonal. After being filtered, each methanolic solution was injected directly into UPLC system. Meanwhile, 20 μL aliquots of each CG and MCG sample were mixed to obtain a quality control (QC) sample, which contained all of the components in the analysis. The QC sample was run randomly to monitor the stability of the system. All of the above solutions were stored at 4 °C prior to LC-MS analysis and the injection volume was 2 μL. 

### 2.3. UPLC/QTOF-MS^E^

The chromatographic separation and mass spectrometry detection were conducted on the Waters Acquity UPLC system coupled with a Xevo G2-S QTOF mass spectrometer equipped with an electrospray ionization source (ESI). Separation was performed on Waters ACQUITY UPLC BEH C_18_ column (100 mm × 2.1 mm, 1.7 μm) at 40 °C. The mobile phase consisted of eluent A (0.1% formic acid aqueous solution) and eluent B (0.1% formic acid in acetonitrile) at flow rate of 0.4 mL/min with the following gradient program: 0~2 min, 10% (B); 2~26 min, 10%~100% (B); 26~28 min, 100% (B); 28~28.1 min, 100%~10% (B); 28.1~30 min, 10% (B). Mixtures of 10/90 and 90/10 water/acetonitrile were the strong wash and the weak wash solvent, respectively. The optimized conditions were employed: source temperature was 120 °C, the desolvation temperature was 300 °C, capillary voltage was 2.6 kV(ESI^+^) or 2.2 kV (ESI^−^), cone voltage was 40 V, desolvation gas flow was 800.0 L/h, cone gas flow was 50 L/h. The energy of low energy function and the collision energy of high energy function were set at 6 V and 20 V~40 V respectively in MS^E^ mode. The mass spectrometer was calibrated with sodium formate in the range of 200–1500 Da. The lockmass compound used was leucine- enkephaline (external reference to the ion *m*/*z* 556.2771 in positive mode and 554.2615 in negative mode). Data were collected with Masslynx™ V4.1 workstation in continuum mode. 

### 2.4. Chemical Information Database for the Components of CG and MCG

In addition to the Waters Traditional Medicine Library in UNIFI software, a systematic investigation of chemical constituents from the target herbs based on the literature was conducted. A self-built database of compounds, such as saponins, flavonoids, volatile oil, amino acids and so on, isolated from CG and MCG was established by searching online databases such as China Journals of Full-Text Database (CNKI), PubMed, Medicine, Web of Science and ChemSpider. The name, molecular formula and structure of components from CG and MCG were obtained in the database. 

### 2.5. The Screening Analysis Based on UNIFI Platform 

UNIFI 1.7.0 software (Waters, Manchester, UK) was used to perform the screening analysis on the structural characteristics and MS fragmentation behaviors, especially for characteristic fragments. Main parameters were set as follows: peak intensity of high energy over 200 counts and the peak intensity of low energy over 1000 counts were the selected parameters in peak detection; mass error up to ±10 ppm for identified compound; retention time tolerance was set in the range of ±0.1 min; positive adducts containing +H, +Na or negative adducts containing −H, +HCOOH were all selected; the reference compound was leucine-enkephalin (556.2766 for positive ion, 554.2620 for negative ion). The MS raw data were processed using the streamlined workflow of UNIFI software to quickly identify the chemical components that met the match criteria with the in-house Traditional Medicine Library and the self-built database [20,21]. 

### 2.6. The Metabolomics Analysis Based on Multivariate Statistical Analysis

To differentiate MCG and CG, MarkerLynx XS V4.1 software (Waters, Milford, DE, USA) was used to process the raw data by deconvolution, alignment, data reduction and to perform the multivariate statistical analysis [20,21]. The following steps were performed: acquiring data, creating a MarkerLynx processing method, processing the acquired data and viewing results Extended Statistics (XS) Viewer. The main parameters in the method set to process the raw data were as follows: retention time range 5–28 min, mass range 200–1400 Da, mass tolerance 5 mDa, intensity threshold 2000 counts, mass window 0.05 Da, retention time window 0.20 min. In resulting database list, RT-*m*/*z* pairs represent an identifier of ion in the order of their elution time. The same value of RT and *m*/*z* in different batches of samples were regarded as the same compound. Multivariate statistical analysis was then performed to find the potential biomarkers that significantly contributed to the difference among the groups. During the analysis, principal component analysis (PCA) was firstly used to show the maximum variation and pattern recognition in order to get the overview and classification, and the orthogonal projections to latent structures discriminant analysis (OPLS-DA) was then performed aiming to get the maximum separation between two groups. S-plots was then available to provide visualization of the OPLS-DA predictive component loading to facilitate model interpretation. Variable importance for the projection (VIP) was also used to help screen the different components, and the metabolites with VIP value above 1.0 were considered as potential markers. Additionally, a permutation test was performed to provide reference distributions of the R^2^/Q^2^-values that could indicate the statistical significance. Simca 15.0 software (Umetrics, Malmö, Sweden) was used to show the analysis results. 

## 3. Results and Discussion

### 3.1. Identification of Components from MCG and CG Based on UNIFI Platform

As a result of our analysis, a total of 126 compounds, including triterpenoids (the main ingredients), flavonoids, organic acids and organic acid esters, alcohol phenols, aldehyde ketones and amino acids, etc., were characterized or tentatively identified from the MCG and CG in both ESI^+^ and ESI^−^ modes. 85 compounds were identified in ESI^+^ mode and 41 compounds were identified in ESI^-^ mode. Base peak intensity (BPI) chromatograms are shown in Figure 1, the identification information is listed in Table 2, and the chemical structures are shown in Figure 2. 

For the isomers, they could be compared with the retention time of the standards or distinguished by the characteristic MS fragmentation patterns reported in literature. Taking compounds **82** and **88** as example, both of them had the same protonated ion [M + HCOO]^−^ at *m*/*z* 991.5464 and 991.5476. In a result, one of them was identified as ginsenoside Rd due to the same retention time, and the other one was tentatively identified as gypenoside XVII because it was matched with the characteristic MS fragmentation pattern of gypenoside XVII reported in the literature [31].

### 3.2. Biomarker Discovery for Distinguishing MCG and CG

The MS^E^ data of CG and MCG samples were statistically analyzed via PCA and OPLS-DA. As seen in PCA 2D plots (Figure 3), there was no obvious difference among of 4–6-year-old CG samples, but the MCG_20 years_, MCG_12 years_ and CG_4–6 years_ groups were obviously separated, indicating that these three groups could be differentiated. With the aim of distinguishing MCG from CG, or MCG_20 years_ from MCG_12 years_, OPLS-DA plot, permutation test, and S-plot, VIP values were obtained to understand which variables were responsible for the separation (Figure 4, Figure 5 and Figure 6). The variables showing VIP > 1 and *p* < 0.05 (in *t*-test) were considered as potential biomarkers. The robust known biomarkers enabling the differentiation between CG and MCG were discovered and marked in S-plots. In order to systematically evaluate the biomarkers, heatmaps (Figure 7) were generated from these biomarkers. The hierarchical clustering heatmaps, intuitively visualizing the differential levels of potential biomarkers concentration in different ginseng groups, are shown in Figure 7. The larger contents were represented by red squares and smaller values by green squares. 

Between the CG_4–6 years_ and MCG_12 years_ groups, the contents of 24-hydroxyoleanolic acid, ginsenoside F_3_ and palmitoleic acid in MCG_12_ samples were significantly higher. While, the contents of α-linolenic acid, 9-octadecenoic acid, linoleic acid and panaxydol in all the CG samples were significantly higher. 

Between the CG_4–6 years_ and MCG_20 years_ groups, the contents of ginsenoside Re_1_, -Re_2_, -Rs_1_, malonylginsenoside Rb_2_, -Rf, isomer of malonylginsenoside-Rb_1_ and quinquenoside R_1_ in the samples of MCG_20 years_ were higher. On the contrary, the contents of ginsenoside Ro and the isomer of ginsenoside Ro, 12,13,15-trihydroxy-9-octadecenoic acid, linoleic acid, 9-octadecenoic acid, α-linolenic acid, panaxydol were rather higher in CG samples. 

Between the MCG_12 years_ and MCG_20 years_ groups, the contents of palmitoleic acid and 24-hydroxyoleanolic acid in MCG_12 years_ samples were significantly high, while the contents of ginsenoside Re_1_, -Rs_1_, malonylginsenoside Rb_2_, -Re_2_ and isomer of malonylginsenoside Rb_1_ were rather higher in MCG_20 years_ samples. 

Overall, on one hand, the contents of α-linolenic acid, linoleic acid, 9-octadecenoic acid and panaxydol in CG samples were significantly higher than those in all MCG samples. On the other hand, ginsenoside Re_1_, -Re_2_, -Rs_1_, malonylginsenoside Rb_2_ and isomer of malonylginsenoside Rb_1_ in MCG_20 years_ samples were really higher than those both in MCG_12 years_ and in all of CG samples, but there is no significant difference between MCG_12 years_ and CG_4–6 years_ samples. The summary with variable identity, VIP and *p* value were shown in Table 3. 

## 4. Discussion

Although MCG and CG both belong to *Panax ginseng,* their chemical ingredients and pharmacological activities are different due to their significantly different growth environment [3,67]. As we all know, MCG has been regarded as a replacement of wild ginseng. Recently, the UPLC-QTOF-MS/MS-based approach has been developed to distinguish MCG (grown for 15 years) and CG (grown for 4–7 years) [6]. As a result, 40 ginsenosides in both MCG and CG were unambiguously identified and tentatively assigned, and the potential chemical markers identifying different ginseng products were characterised [6]. Additionally, the study on 6–18-year-old Mountain Cultivated Ginseng Leaves (MGL) samples showed that the MGL were obviously divided into three main groups according to different age brackets (6~10, 11~13 and 14~18 years) [7]. Although the sample of the study was the leaf of MCG, it could be indirectly speculated that the MCG roots with different cultivation ages are also different. In order to further systematically compare the similarities and differences at the chemical level between different ages of ginseng, especially to compare the younger or the older MCG, 4, 5, 6-year-old CG and 12, 20-year-old MCG were chosen as the analytical samples in the present study.

Firstly, based on UNIFI platform, intelligent and automatic workflows, the screening analysis of metabolites in different cultivation ages of ginseng were rapidly performed. As a result, a total of 126 compounds were characterized from CG_4–6 years_, MCG_12 years_ and MCG_20 years_ samples. Among of them, ginsenosides were the main ingredients. Both CG and MCG had the similar chemical composition, but the components were variously distributed in CG and MCG samples at different contents. That means in CG and MCG, the secondary metabolites had the features of structural diversity and the different content patterns. As far as we know, this is the first time that the comprehensive screening analysis of MCG_12 years_ and MCG_20 years_ samples by using UPLC-QTOF-MS^E^ combined with UNIFI platform. It could provide the scientific data for clarifying the chemical composition of MCG. 

Secondly, the combination of LC-MS based metabolomic profiling with multivariate statistical analysis method was used to profile the CG, MCG_12 years_ and MCG_20 years_ samples. A total of 17 potential age-dependent markers enabling differentiation among the CG and MCG samples were discovered. (1) There were four robust markers including α-linolenic acid, 9-octadecenoic acid, linoleic acid and panaxydol being the characteristic components for CG samples, that distinguished them from both MCG_12 years_ and MCG_20 years_ samples. The results showed that CG samples contained more non-ginsenosides. Both linoleic acid and α-linolenic acid, the main products of the acetate-malonate pathway, are two essential fatty acids necessary for health. Linoleic acid is used in the biosynthesis of arachidonic acid and thus some prostaglandins, leukotrienes, and thromboxane [68,69]. Panaxydol, one of the C17 polyacetylenic compounds, originates from acetyl-CoA/malonyl-CoA via fatty acids with crepenynate as the intermediate [70]. It is considered a potential antitumor agent due to its significant anticancer activity [71]. (2) In CG samples, there were three other characteristic components such as ginsenoside Ro, the isomer of ginsenoside Ro, and 12,13,15-trihydroxy-9-octadecenoic acid, that could be used to differentiate them from MCG_20 years_ samples. From this, we could draw a conclusion that pentacyclic triterpenoids decreased significantly in older MCG samples. (3) Five robust biomarkers including ginsenoside Re_1_, -Re_2_, -Rs_1_, malonylginsenoside Rb_2_ and isomer of malonylginsenoside Rb_1_ were found to enable differentiation of MCG_20 years_ from CG and MCG_12 years_ samples. These five compounds might be used for rapid identification of MCG_20 years_ samples. A proposed biosynthetic pathway of ginsenosides is as follows: with the action of squalene epoxidase, squalene was converted to 2,3-oxidosqualene. Dammaranes can be synthesized by dammarenediol synthase, and oleananes by β-amyrin synthase [72]. Ginsenosides were found to have both antimicrobial and antifungal properties and the molecules are naturally bitter-tasting, discouraging insects and other animals from consuming the plant, so ginsenosides likely serve as mechanisms for plant defense [73,74]. (4) In MCG_20 years_ samples, another two markers, ginsenoside Rf and quinquenoside R_1_, were discovered that distinguished them from all CG samples. (5) In MCG_12 years_ samples, 24-hydroxyoleanolic acid and palmitoleic acid were the two robust markers for distinguished from both CG and MCG_20 years_ samples. These two compounds might be used for rapid identification of MCG_12 years_ samples. Palmitoleic acid is biosynthesized from palmitic acid by the action of the enzyme stearoyl-CoA desaturase-1, a key enzyme in fatty acid metabolism [75]. (6) Ginsenoside F_3_ was another marker for MCG_12 years_ samples that differentiated them from CG samples. However, there are still some unresolved issues. For example, as shown in BPI chromatograms, though 126 compounds were identified, there are still some unidentified components. there are still some unidentified components. Further research should be carried out based on the formula of these unknown compounds. 

## 5. Conclusions

By combining the UPLC-Q/TOF-MS^E^ and UNIFI platform, 126 chemical components with various structural types, such as triterpenoids, flavonoids, organic acids and organic acid esters, etc., were characterized or tentatively identified from CG_4–6 years_, MCG_12 years_ and MCG_20 years_ samples for the first time. All the CG and MCG samples had the similar chemical composition, but there were significant differences in the content of each component. Further nontarget metabolomic analysis combined with multivariate statistical analysis showed that CG_4–6 years_, MCG_12 years_ and MCG_20 years_ samples were obviously divided into three different groups. A total of 17 potential age-dependent markers enabling differentiation among the CG and MCG samples were discovered. Among of these markers, four robust markers, including α-linolenic acid, 9-octadecenoic acid, linoleic acid and panaxydol, could be the characteristic components for differentiation of CG from all other MCG samples. Five robust markers including ginsenoside Re_1_, -Re_2_, -Rs_1_, malonylginsenoside Rb_2_ and isomer of malonylginsenoside Rb_1_ were found to enable differentiate MCG_20 years_ samples from all other samples, while 24-hydroxyoleanolic acid and palmitoleic acid were the robust markers for distinguishing MCG_12 years_ samples from all the CG samples and MCG_20 years_ samples. The proposed approach could be applied to directly distinguish MCG root ages, which is an important criterion for evaluating the quality of MCG. The results will provide the data for the deficient study on the chemical constituents of MCG and provide reference for the quantitative determination in the quality control criterion of MCG.

## Figures and Tables

**Figure 1 molecules-24-00033-f001:**
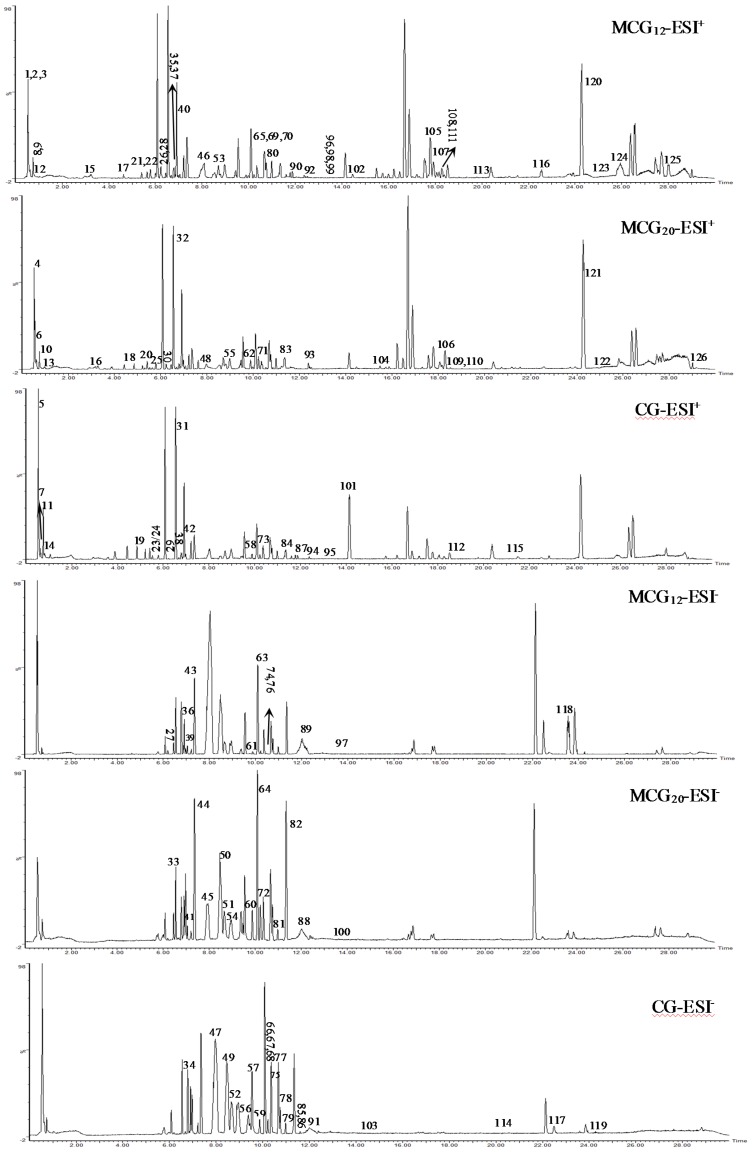
The representative BPI chromatograms of CG and MCG in positive and negative modes.

**Figure 2 molecules-24-00033-f002:**
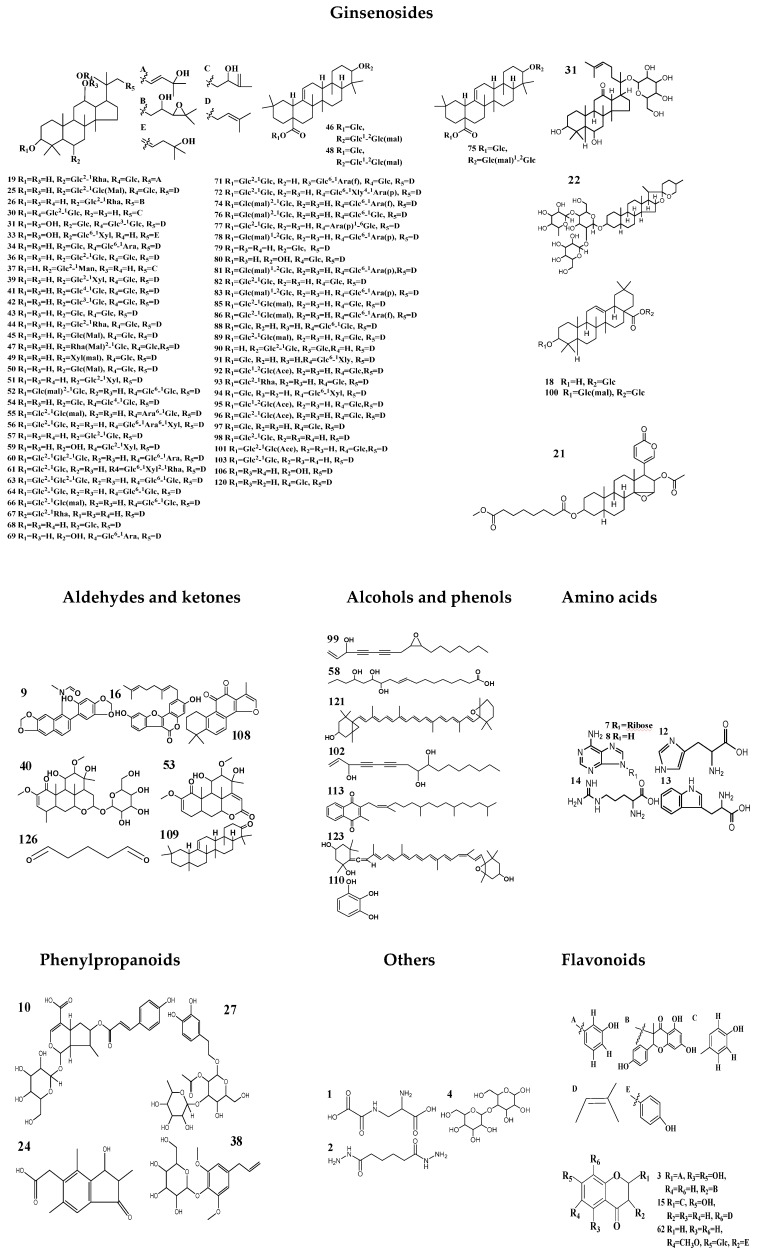
Chemical structures of compounds identified in MCG and CG.

**Figure 3 molecules-24-00033-f003:**
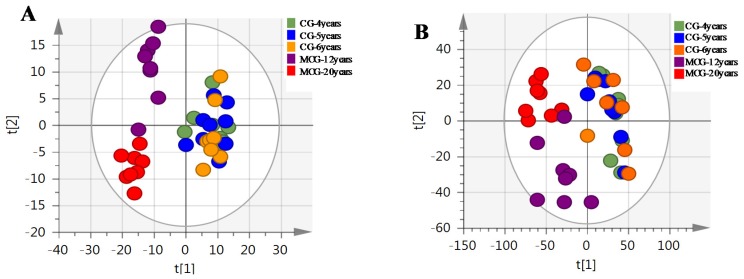
The PCA of CG and MCG in positive mode (**A**) and negative mode (**B**).

**Figure 4 molecules-24-00033-f004:**
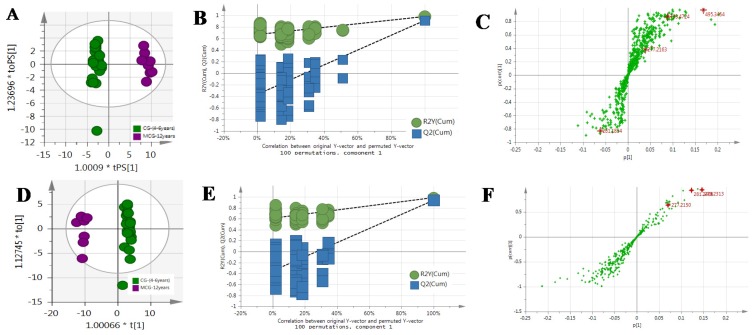
The OPLS-DA/Permutation test/S-Plot of CG_4–6 years_ and MCG_12 years_ in positive mode (**A**/**B**/**C**) and negative mode (**D**/**E**/**F**).

**Figure 5 molecules-24-00033-f005:**
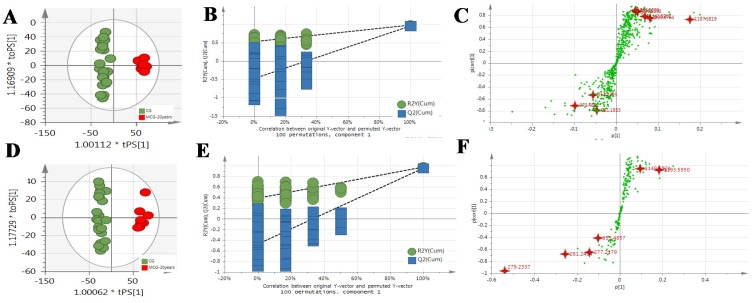
The OPLS-DA/Permutation test/S-Plot of CG_4–6 years_ and MCG_20 years._ in positive mode (**A**/**B**/**C**) and negative mode (**D**/**E**/**F**).

**Figure 6 molecules-24-00033-f006:**
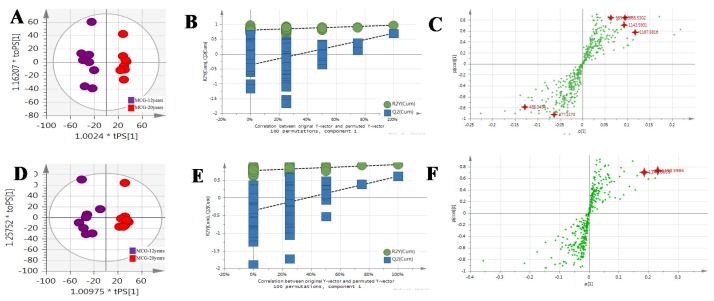
The OPLS-DA/Permutation test/S-Plot of MCG_12 years_ and MCG_20 years_ in positive mode (**A**/**B**/**C**) and negative mode (**D**/**E**/**F**).

**Figure 7 molecules-24-00033-f007:**
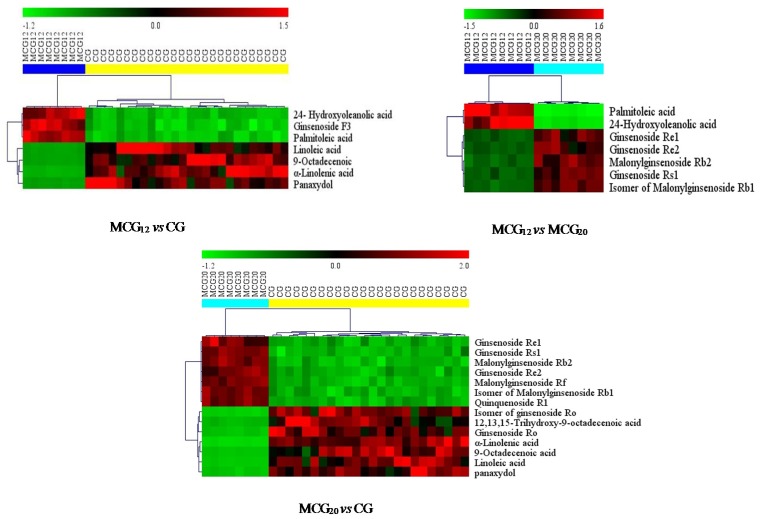
The heatmaps visualizing the intensities of potential biomarkers.

**Table 1 molecules-24-00033-t001:** Details of the MCG and CG samples.

Sample No.	Source	Collection Time
CG_3years_-1, CG_3years_-2; CG_4years_-1, CG_4years_-2; CG_5years_-1, CG_5years_-2; MCG_12years_-1, MCG_12years_-2; MCG_20years_-1, MCG_20years_-2	Ji′an City, Jilin Province, China	2017.09–2017.10
CG_3years_-3, CG_3years_-4; CG_4years_-3, CG_4years_-4; CG_5years_-3, CG_5years_-4; MCG_12years_-3, MCG_12years_-4; MCG_20years_-3, MCG_20years_-4	Fusong County, Jilin Province, China	2017.09–2017.10
CG_3years_-5, CG_3years_-6; CG_4years_-5, CG_4years_-6; CG_5years_-5, CG_5years_-6; MCG_12years_-5, MCG_12years_-6; MCG_20years_-5, MCG_20years_-6	Tonghua City, Jilin Province, China	2017.09–2017.10
CG_3years_-7, CG_3years_-8; CG_4years_-7, CG_4yeasr_-8; CG_5years_-7, CG_5years_-8; MCG_12years_-7, MCG_12years_-8; MCG_20years_-7, MCG_20years_-8	Jingyu Country, Jilin Province, China	2017.09–2017.10

**Table 2 molecules-24-00033-t002:** Compounds identified from MCG and CG by UPLC-QTOF-MS^E^.

No.	t_R_ (min)	Formula	Calculated Mass (Da)	Theoretical Mass (Da)	Mass Error (ppm)	MS^E^ Fragmentation	Identification	Sources	Ref.
1	0.49	C_5_H_8_N_2_O_5_	176.0431	176.0433	−1.5	177.0503[M + H]^+^; 130.0495[M – 2 × OH − NH_2_]^+^	Dencichine	CG, MCG_12_, MCG_20_	[23]
2	0.54	C_6_H_14_N_4_O_2_	174.1115	174.1117	−1.1	175.1188[M + H]^+^;158.0912[M − NH_2_]^+^; 116.0704[M − NH_2_ − CN_2_H_2_]^+^;114.1015[M − NH_2_ − CHO_2_]^+^	Adipodihydrazide	CG, MCG_12_, MCG_20_	a
3	0.55	C_30_H_22_O_10_	542.1257	542.1213	8.1	543.1330[M + H]^+^; 273.0833[M − C_15_H_10_O_5_]^+^; 242.1025[M − OH − C_15_H_10_O_6_]^+^; 127.0388[M − C_24_H_17_O_7_]^+^; 116.0703[M − 2 × OH − C_21_H_12_O_8_]^+^; 109.0284[M − C_24_H_15_O_8_]^+^	Chamaejasmine	CG, MCG_12_, MCG_20_	a
4	0.59	C_12_H_22_O_11_	342.1156	342.1162	−1.6	365.1055[M + Na]^+^; 203.0550[M − OH − C_4_H_8_O_4_]^+^; 185.0444[M − 2 × OH − C_4_H_8_O_4_]^+^	α-Maltose	CG, MCG_12_, MCG_20_	[24]
5	0.69	C_6_H_14_N_4_O_2_	174.1115	174.1117	−0.9	175.1188[M + H]^+^	d-Arginin	CG, MCG_12_, MCG_20_	s
6	0.74	C_19_H_18_O_11_	422.0842	422.0849	−1.7	423.0915[M + H]^+^; 268.1040[M + H − C_6_H_6_O_5_]^+^; 119.0349[M − C_15_H_10_O_7_]^+^	Isomangiferin	CG, MCG_12_, MCG_20_	[25]
7	0.75	C_5_H_5_N_5_	135.0546	135.0545	0.5	136.0618[M + H]^+^; 119.0352[M − NH_2_]^+^	Adenine	CG, MCG_12_, MCG_20_	s
8	0.76	C_10_H_13_N_5_O_4_	267.0974	267.0968	2.4	268.1050[M + H]^+^; 136.0618[M − C_5_H_9_O_4_]^+^; 119.0352[M − C_5_H_9_O_4_ − NH_2_]^+^	d-Adenosine	CG, MCG_12_, MCG_20_	s
9	0.80	C_20_H_15_NO_6_	365.0876	365.0899	−6.3	366.0949[M + H]^+^	Integriamide	CG, MCG_12_, MCG_20_	a
10	0.81	C_25_H_30_O_12_	538.1677	538.1686	−1.7	561.1569[M + Na]^+^; 393.1138[M − C_6_H_8_O_4_]^+^; 381.0788[M − CH_3_ − C_7_H_10_O_3_]^+^; 366.0930[M − OH − C_8_H_10_O_3_]^+^; 366.0930[M − C_19_H_21_O_8_]^+^	Linearoside	CG, MCG_12_, MCG_20_	[26]
11	0.82	C_9_H_11_NO_2_	165.0782	165.0790	−0.5	166.0862[M + H]^+^; 120.0805[M − COOH]^+^_;_ 103.0543[M − COOH − NH_2_]^+^	Phenylpropionic acid	CG, MCG_12_, MCG_20_	s
12	0.91	C_6_H_9_N_3_O_2_	155.0762	155.0695	−3.5	156.0762[M + H]^+^	Histidine	CG, MCG_12_, MCG_20_	s
13	1.05	C_11_H_12_N_2_O_2_	204.0898	204.0899	−0.5	205.0971[M + H]^+^; 188.0706[M − NH_2_]^+^; 143.0723[M − NH_2_ − COOH]^+^; 118.0649[M − NH_2_ − COOH − C_2_H_3_]^+^	Tryptophan	CG, MCG_12_, MCG_20_	s
14	1.06	C_6_H_14_N_4_O_2_	174.1117	174.1117	0.0	175.1190[M + H]^+^	Argentine	CG, MCG_12_, MCG_20_	[27]
15	3.13	C_25_H_28_O_4_	392.2009	392.1988	5.5	393.2082[M + H]^+^	Glabrol	CG, MCG_12_, MCG_20_	[28]
16	3.27	C_25_H_24_O_5_	404.1646	404.1624	5.1	405.1719[M + H]^+^	Puerarol	CG, MCG_12_, MCG_20_	[29]
17	4.42	C_27_H_38_O_6_	458.2716	458.2668	10.0	481.2609[M + Na]^+^_;_ 436.2642[M − COOH]^+^	Lucideric acid	CG, MCG_12_, MCG_20_	a
18	4.64	C_36_H_58_O_8_	618.4107	618.4132	−4.0	619.4180[M + H]^+^; 421.3446[M − Glc − OH]^+^	β-d-Glcopyranosyl oleanolate	CG, MCG_12_, MCG_20_	[30]
19	4.89	C_48_H_82_O_19_	962.5484	962.5450	3.3	985.5312[M + Na]^+^; 765.4795[M − Glc − OH]^+^; 541.2637[M − Glc − OH − C_15_H_28_O]^+^; 421.3463[M − Glc − Glc/Rha − 2 × OH]^+^	Majoroside F6	CG, MCG_12_, MCG_20_	[31]
20	5.21	C_31_H_46_O_8_	546.3248	546.3193	9.7	569.3140[M + Na]^+^; 133.0859[M − C_25_H_33_O_5_]^+^	Methyl ganoderate G	CG, MCG_12_, MCG_20_	a
21	5.53	C_35_H_48_O_9_	612.3344	612.3298	7.4	613.3416[M + H]^+^_;_ 582.3264[M − OCH_3_]^+^_;_ 526.2986[M − C_4_H_7_O_2_]^+^	Cinobufagin 3-hemisuberate methyl ester	CG, MCG_12_, MCG_20_	a
22	5.56	C_45_H_74_O_17_	886.4877	886.4926	−5.4	909.4769[M + Na]^+^; 745.4383[M − CH_2_OH − C_6_H_10_O_2_]^+^; 729.4136[M − OH − C_8_H_14_O_2_]^+^; 601.2768[M − 2 × OH − C_16_H_26_O_4_]^+^; 431.1870[M − Glc − C_19_H_27_O_2_]^+^	Shatavarin IV	CG, MCG_12_, MCG_20_	[32]
23	5.64	C_25_H_32_O_13_	540.1800	540.1843	3.0	541.1873[M + H]^+^; 347.0906[M − CH_3_O − C_2_H_4_ − C_8_H_9_O_2_]^+^; 195.1008[M − Glc − C_2_H_4_O_2_ − C_6_H_5_O_2_]^+^	Oleuropein	CG, MCG_12_, MCG_20_	[33]
24	5.65	C_14_H_16_O_4_	248.1025	248.1049	−9.7	271.0917[M + Na]^+^; 195.1008[M − C_2_H_4_ − COH]^+^; 189.1348[M − OH − COOH]^+^	Isohistiopterosin A	CG, MCG_12_, MCG_20_	a
25	5.79	C_45_H_74_O_17_	886.4881	886.4926	−5.0	909.4773[M + Na]^+^; 707.4360[M − Glc]^+^; 689.4262[M − Glc − OH]^+^; 657.3636[M − Glc − OH − 2 × CH_3_]^+^; 609.3646[M − Glc − C_6_H_10_O_2_]^+^; 523.3626[M − Glc/Glc − C_3_H_6_]^+^	Malonylginsenoside Rf	CG, MCG_12_, ^##^ MCG_20_	[34]
26	6.21	C_42_H_70_O_14_	798.4778	798.4766	1.6	799.4851[M + H]^+^; 439.3563[M − Glc/Rha − 2 × OH]^+^; 421.3441[M − Glc/Rha − 3 × OH − H_2_O]^+^	Ginsenoside Rg_8_	CG, MCG_12_, MCG_20_	[34]
27	6.23	C_22_H_32_O_13_	504.1840	504.1843	−0.6	503.1767[M − H]^−^; 457.1715[M − OH − CH_2_OH]^−^; 293.0878[M − C_2_H_4_O − CH_2_OH − C_8_H_9_O_2_]^−^	Cistanoside H	CG, MCG_12_, MCG_20_	[35]
28	6.24	C_23_H_28_O_11_	480.1594	480.1632	−7.7	481.1667[M + H]^+^; 317.0803[M − C_10_H_12_O_2_]^+^	Peoniflorin	CG, MCG_12_, MCG_20_	a
29	6.28	C_24_H_30_O_12_	510.1702	510.1737	−6.9	511.1775[M + H]^+^; 317.0803[M − C_11_H_14_O_3_]^+^	Mudanpioside D	CG, MCG_12_, MCG_20_	[36]
30	6.39	C_54_H_92_O_24_	1124.5943	1124.5978	−3.1	1147.5835[M + Na]^+^; 585.2870[M − C_25_H_47_O_12_]^+^; 325.1130[M − C_42_H_71_O_14_]^+^	Ginsenoside V	CG, MCG_12_, MCG_20_	[31]
31	6.49	C_48_H_82_O_19_	962.5484	962.5450	3.3	985.5302[M + Na]^+^; 865.4789[M − C_6_H_9_O]^+^; 823.4787[M − C_8_H_11_O_2_]^+^; 805.4668[M − C_8_H_13_O_3_]^+^; 555.2763[M − C_12_H_26_O_5_]^+^; 423.3602[M − Glc − Glc/Glc − OH]^+^; 405.3507[M − Glc − Glc/Glc − 2 × OH]^+^	Ginsenoside Re_1_	CG, MCG_12_, ^##,^* MCG_20_	[37]
32	6.59	C_23_H_28_O_11_	480.1587	480.1632	−9.3	481.1660[M + H]^+^; 317.0810[M − C_7_H_5_O − C_3_H_5_O]^+^	Mudanpioside I	CG, MCG_12_, MCG_20_	[38]
33	6.64	C_41_H_70_O_14_	786.4762	786.4766	−0.4	831.4744[M + HCOO]^−^; 653.4270[M − H − C_5_H_8_O_4_]^−^, 491.3710[M − H − C_11_H_18_O_9_]^−^	Notoginsenoside Rw2	CG, MCG_12_, MCG_20_	[39]
34	6.67	C_47_H_80_O_18_	932.5335	932.5345	−1.0	977.5317[M + HCOO]^−^; 785.4693[M − Ara − CH_3_]^−^;653.4282[M − Glc − 2 × OH − C_5_H_9_]^−^	Quinquenoside F_6_	CG, MCG_12_, MCG_20_	[37]
35	6.77	C_36_H_60_O_9_	636.4217	636.4237	−3.1	637.4290[M + H]^+^; 621.42740[M − OH]^+^; 423.3605[M − Glc − 2 × OH]^+^	Ginsenoside Rh_8_	CG, MCG_12_, MCG_20_	[40]
36	6.84	C_48_H_82_O_19_	962.5469	962.5450	1.9	1007.5456[M + HCOO]^−^; 799.4848[M − Glc]^−^; 637.4317[M − Glc/Glc]^−^; 179.0545[Glc − H]^−^	20-β-d-Glucopyranosyl-ginsenoside Rf	CG, MCG_12_, MCG_20_	[41]
37	6.80	C_42_H_70_O_13_	782.4773	782.4816	−5.4	805.4665[M + Na]^+^; 765.4734[M − OH]^+^; 677.4220[M − 2 × OH − C_4_H_7_O]^+^; 661.4265[M − 3 × OH − C_4_H_7_O]^+^; 439.3562[M − Glc − Man − OH]^+^	Ginsenoside Rh_14_	CG, MCG_12_, MCG_20_	[40]
38	6.82	C_17_H_24_O_8_	356.1460	356.1472	−3.1	379.1352[M + Na]^+^; 145.0495[M − OH − C_11_H_13_O_3_]^+^	Erigeside II	CG, MCG_12_, MCG_20_	[42]
39	6.96	C_47_H_80_O_18_	932.5410	932.5345	6.7	977.5392[M + HCOO]^−^; 799.4825[M − Xyl]^−^; 769.4724[M − H − Glc]^−^; 637.4291[M − (Glc/Xyl) ]^−^; 179.0539 [Glc − H]^−^	Notoginsenoside R_1_	CG, MCG_12_, MCG_20_	s
40	6.99	C_28_H_44_O_12_	572.2810	572.2833	−3.9	573.2883[M + H]^+^; 555.2779[M − OH]^+^; 531.2860[M − C_2_H_3_O]^+^	Picrasinoside G	CG, MCG_12_, MCG_20_	a
41	7.05	C_48_H_82_O_19_	962.5425	962.5450	−2.6	1007.5415[M + HCOO]^−^; 799.4822[M − Glc]^−^; 637.4333[M − (Glc/Glc) ]^−^	Notoginsenoside N	CG, MCG_12_, MCG_20_	[43]
42	7.20	C_48_H_82_O_19_	962.5422	962.5450	−2.9	985.5314[M + Na]^+^; 703.4371[M − Glc − 2 × OH − CH_2_OH]^−^; 439.3565[M − Glc − Glc/Glc − OH]^−^	Ginsenoside Re_2_	CG, MCG_12_, ^##,^**MCG_20_	[40]
43	7.34	C_42_H_72_O_14_	800.4934	800.4922	1.4	845.4916[M + HCOO]^−^; 637.4344[M − Glc]^−^; 475.3798[M − Glc − Glc]^−^; 179.0553[Glc − H]^−^;	Ginsenoside Rg_1_	CG, MCG_12_, MCG_20_	s
44	7.36	C_48_H_82_O_18_	946.5524	946.5501	2.3	991.5506[M + HCOO]^−^; 783.4912[M − Glc]^−^; 637.4344[M − (Glc/Rha)]^−^; 475.3798[M − Glc − (Glc/Rha)]^−^	Ginsenoside Re	CG, MCG_12_, MCG_20_	s
45	7.74	C_45_H_74_O_17_	886.4925	886.4926	−0.1	885.4853[M − H]^−^; 781.4740[M − HOCOCH_2_COOH]^−^; 619.4197[M − Glc(Mal)]^−^; 161.0438[Glc − H_2_O]^−^	Malonylginsenoside Rg_1_	CG, MCG_12_, MCG_20_	[39]
46	7.93	C_48_H_76_O_19_	956.4960	956.4981	−2.2	979.4852[M + Na]^+^; 799.4161[M − CO_2_ − CH_2_OH − C_6_H_12_]^+^; 641.4008[M − Glc − C_4_H_6_O_5_]^+^;439.3562[M − Glc − Glc/Glc(mal)]^+^; 145.0493[Glc − OH]^+^	Isomer of ginsenoside Ro	^#^ CG, MCG_12_, MCG_20_	[31]
47	8.04	C_51_H_84_O_21_	1032.5532	1032.5505	2.6	1031.5460[M − H]^−^; 987.5564[M − CO_2_]^−^; 927.5337[M − HOCOCH_2_COOH]^−^; 781.4759[M − Rha(Mal) ]^−^; 619.4222[M − (Rha(Mal)/Glc]^−^	Malonylginsenoside Re	CG, MCG_12_, MCG_20_	[39]
48	8.08	C_48_H_76_O_19_	956.4950	956.4981	−3.1	979.4842[M + Na]^+^; 817.4311[M − Glc]^+^; 439.3571[M − Glc/Glc − Glc − OH]^+^	Isomer of ginsenoside Ro	CG, MCG_12_, MCG_20_	[44]
49	8.09	C_44_H_74_O_15_	842.5032	842.5028	0.5	841.4959[M − H]^−^; 799.4861[M − CH_2_O]^+^; 781.4741[M − CH_2_O − OH]^+^; 637.4316[M − Xyl(mal)]^+^; 619.4228[M − Xyl(mal) − OH]^+^; 475.3798[M − Xyl(mal) − Glc]^+^; 179.0550[Glc − H]^+^; 161.0439[Glc − OH]^+^	Yesanchinoside D	CG, MCG_12_, MCG_20_	[45]
50	8.10	C_45_H_74_O_17_	886.4931	886.4926	0.6	885.4858[M − H]^−^; 781.4741[M − H − HOCOCH_2_COOH]^−^; 619.4228[M − H − Glc(Mal) ]^−^; 161.0439[Glc − H − H_2_O]^−^	Isomer of malonylginsenoside Rg_1_	CG, MCG_12_, MCG_20_	[39]
51	8.49	C_41_H_70_O_13_	770.4801	770.4816	−1.5	815.4784[M + HCOO]^−^; 637.4321[M − Xyl]^−^	Notoginsenoside R_2_	CG, MCG_12_, MCG_20_	[39]
52	8.50	C_56_H_94_O_24_	1150.6124	1150.6135	−1.1	1149.6051[M − H]^−^; 1119.5951[M − CH_2_OH − 2 × OH]^−^; 807.4861[M − Glc/Glc − OH]^−^; 605.4423[M − Glc/Glc − Glc(mal) ]^−^; 325.1119[Glc/Glc − OH]^−^	Quinquenoside R_1_	CG, MCG_12_, ^##^ MCG_20_	[46]
53	8.60	C_22_H_30_O_47_	406.1957	406.1992	−8.5	407.2030[M + H]^+^; 376.1859[M − OCH_3_]^+^	Nigakilactone K	CG, MCG_12_, MCG_20_	[47]
54	8.87	C_48_H_82_O_19_	962.5445	962.5450	−0.5	1007.5427[M + HCOO]^−^; 797.4706[M − Glc]^−^	Ginsenoside Re_3_	CG, MCG_12_, MCG_20_	[37]
55	8.96	C_56_H_92_O_25_	1164.5929	1164.5928	0.1	1187.5821[M + Na]^+^; 1147.5803[M − OH]^+^; 805.4305[M − Ara/Glc − CH_2_OH − CH_3_]^+^; 443.3868[M − Ara/Glc − Glc/Glc(mal)]^+^	Malonylginsenoside Rb_2_	CG, MCG_12_, ^##,^* MCG_20_	[44]
56	9.41	C_59_H_100_O_27_	1240.6488	1240.6452	2.8	1285.6740[M + HCOO]^−^; 945.5421[M − (Ara/Xyl) ]^−^; 913.5184[M − (Glc/Glc)]^−^; 783.4900[M − (Ara/Xyl) − Glc]^−^	Notoginsenoside R_4_	CG, MCG_12_, MCG_20_	s
57	9.56	C_42_H_72_O_14_	800.4921	800.4922	−0.1	845.4903[M + HCOO]^−^; 637.4319[M − Glc]^−^; 475.3786[M − (Glc/Glc)]^−^; ^1,3^A_2β_221.0658; 161.0439[Glc – H − H_2_O]^−^;^2,5^A_1β_101.0235	Ginsenoside Rf	CG, MCG_12_, MCG_20_	s
58	9.79	C_18_H_34_O_5_	330.2398	330.2406	−2.3	353.2290[M + Na]^+^; 213.1459[M + H – COOH – C_5_H_11_]^+^	12,13,15-Trihydroxy-9-octadecenoic acid	^#^ CG, MCG_12_, MCG_20_	[48]
59	9.87	C_41_H_70_O_13_	770.4809	770.4816	−1.0	815.4791[M + HCOO]^−^; 475.3783[M − (Glc /Xyl)]^−^; 161.0437[Glc – H – H_2_O]^−^	Ginsenoside F_5_	CG, MCG_12_, MCG_20_	s
60	9.89	C_60_H_102_O_28_	1270.6635	1270.6558	5.9	1315.6617[M + HCOO]^−^; 841.4991[M − Glc/Glc – OH – C_4_H_4_]^−^; 769.4777[M − Glc/Glc/Glc – CH_3_]^−^	Ginsenoside Ra_0_	CG, MCG_12_, MCG_20_	[49]
61	9.94	C_58_H_98_O_26_	1210.6358	1210.6346	1.0	1255.6340[M + HCOO]^−^; 1077.5833[M – Xyl]^−^; 1047.5719[M – Glc]^−^; 955.4871[M – Glc – OH – C_4_H_7_]^−^; 783.4892[M – Glc/Xyl/Rha]^−^	Ginsenoside Ra_2_	CG, MCG_12_, MCG_20_	[50]
62	10.00	C_22_H_22_O_10_	446.1192	446.1213	−4.5	469.1084[M + Na]^+^; 429.1154[M – OH]^+^; 385.0884[M – OH – CH_3_ – CH_2_OH]^+^; 341.0661[M – C_4_H_8_O_3_]^+^; 237.0746[M – C_10_H_13_O_5_]^+^; 193.0483[M – C_12_H_16_O_6_]^+^	Glycitin	CG, MCG_12_, MCG_20_	a
63	10.01	C_59_H_100_O_27_	1240.6462	1240.6452	0.8	1285.6444[M + HCOO]^−^; 1107.5964[M-Xyl]^−^; 945.5424[M – (Glc/Xyl)]^−^; 783.4912[M – Xyl – GlcGlc]^−^	Notoginsenoside Fa	CG, MCG_12_, MCG_20_	[50]
64	10.05	C_54_H_92_O_23_	1108.6101	1108.6029	6.2	1153.6083[M + HCOO]^−^; 945.5437[M – Glc]^−^; 783.4888[M – (Glc/Glc)]^−^; 621.4382[M – (Glc/Glc) – Glc]^−^; 459.3835[M – (Glc/Glc) – (Glc/Glc)]^−^; ^2,5^A_1β_101.0235	Ginsenoside Rb_1_	CG, MCG_12_, MCG_20_	s
65	10.10	C_42_H_70_O_12_	766.4863	766.4867	−0.5	767.4936[M + H]^+^; 443.3866[M – Rha – Glc]^+^; 425.3762[M – Rha – Glc – OH]^+^	Ginsenoside Rg_4_	CG, MCG_12_, MCG_20_	[34]
66	10.20	C_57_H_94_O_26_	1194.6087	1194.6033	4.5	1193.6015[M – H]^−^; 1149.6098[M – CO_2_]^−^; 783.4908[M – Glc/Glc)]^−^; 179.0545[Glc – H]^−^	Isomer of malonylginsenoside Rb_1_	CG, MCG_12_, ^##,^** MCG_20_	[39]
67	10.22	C_42_H_72_O_13_	784.4997	784.4973	2.9	829.4979[M + HCOO]^−^; 637.4336[M − Rha]^−^; 475.3809[M – (Glc/Rha)]^−^; 161.0449 [Rha – H]^−^	20(R)-Ginsenoside Rg_2_	CG, MCG_12_, MCG_20_	s
68	10.25	C_36_H_62_O_9_	638.4407	638.4394	2.9	683.4389[M + HCOO]^−^; 161.0449[Glc − H − H_2_O]^−^	Ginsenoside Rh_1_	CG, MCG_12_, MCG_20_	s
69	10.27	C_41_H_70_O_13_	770.4779	770.4816	−4.7	793.4672[M + Na]^+^; 587.4276[M − Ara(p) − 2 × OH]^+^; 423.3589[M − Ara(p)/Glc − 2 × OH]^+^	Ginsenoside F_3_	CG, ^∆∆^ MCG_12_, MCG_20_	[34]
70	10.29	C_36_H_60_O_8_	620.4292	620.4288	0.7	621.4365[M + H]^+^; 390.2277[M − C_17_H_26_]^+^; 187.1473[M − OH − Glc − C_16_H_24_O]^+^	Ginsenoside Rh_4_	CG, MCG_12_, MCG_20_	[40]
71	10.32	C_53_H_90_O_22_	1078.5939	1078.5924	1.3	1101.5805[M + Na]^+^; 939.5312[M − Glc]^+^; 929.5452[M − Ara(f)]^+^; 789.4784[M − Ara(f) − Glc]^+^	Ginsenoside Rc	CG, MCG_12_, MCG_20_	s
72	10.34	C_58_H_98_O_26_	1210.6356	1210.6346	0.7	1255.6338[M + HCOO]^−^; 1077.5851[M − Xly]^−^; 1047.5702[M − Glc]^−^; 945.5396[M − (Xly/ Ara(p))]^−^; 621.4323[M − (Xly/ Ara(p)/Glc − Glc]^−^	Ginsenoside Ra_1_	CG, MCG_12_, MCG_20_	[50]
73	10.38	C_42_H_70_O_12_	766.4872	766.4867	0.6	767.4945[M + H]^+^;605.4423[M − Glc]^+^;443.3870[M − Glc/Xyl]^+^;407.3660[M − Glc − 2 × OH]^+^; 163.0591[Glc − OH]^+^;145.04901[Glc − OH − H_2_O]^+^	Ginsenoside Rg_5_	CG, MCG_12_, MCG_20_	s
74	10.47	C_56_H_92_O_25_	1164.5947	1164.5928	1.6	1163.5874[M − H]^−^; 1119.5961[M − CO_2_]^−^; 927.5320[M − Ara(f) − HOCOCH_2_COOH]^−^	Malonylginsenoside Rc	CG, MCG_12_, MCG_20_	[44]
75	10.51	C_48_H_76_O_19_	956.5001	956.4981	2.1	955.4928[M − H]^−^; 793.4399[M − Glc]^−^; 613.3739[M − Glc − Glc − OH]^−^	Ginsenoside Ro	^#^ CG, MCG_12_, MCG_20_	s
76	10.57	C_57_H_94_O_26_	1194.6059	1194.6033	2.2	1193.5986[M − H]^−^; 1149.6062[M − CO_2_]^−^; 1089.5851[M − HOCOCH_2_COOH]^−^; 945.5428[M − Glc(Mal)]^−^; 783.4926[M − (Glc/Glc)]^−^	Malonylginsenoside Rb_1_	CG, MCG_12_, MCG_20_	[39]
77	10.63	C_53_H_90_O_22_	1078.5979	1078.5924	4.9	1123.5961[M + HCOO]^−^; 945.5448[M − Ara(p)]^−^; 783.4896[M − (Ara/Glc)]^−^; 149.0443[Ara(p) − H]^−^	Ginsenoside Rb_2_/Rb_3_	CG, MCG_12_, MCG_20_	s
78	10.77	C_56_H_92_O_25_	1164.5986	1164.5928	5.0	1163.5913[M − H]^−^; 1101.5822[M − CO_2_]^−^; 765.4782[M − H − Glc(Mal) − Ara(p) − OH]^−^	Malonylginsenoside Rb_2_	CG, MCG_12_, MCG_20_	[44]
79	11.06	C_36_H_62_O_9_	638.4391	638.4394	−0.4	683.4373[M + HCOO]^−^	20(*R*)-Ginsenoside Rh_1_	CG, MCG_12_, MCG_20_	s
80	11.14	C_36_H_62_O_9_	638.4399	638.4394	0.7	661.4291[M + Na]^+^; 376.2462[M − C_17_H_24_O_2_]^+^	Ginsenoside F_1_	CG, MCG_12_, MCG_20_	s
81	11.15	C_56_H_92_O_25_	1164.5971	1164.5928	3.7	1163.5898[M − H]^−^; 1119.6000[M − CO_2_]^−^; 1059.5772[M − H − C_3_H_4_O_4_]^−^;	Malonylginsenoside Rb_3_	CG, MCG_12_, MCG_20_	[39]
82	11.27	C_48_H_82_O_18_	946.5482	946.5501	−1.9	991.5464[M + HCOO]^−^; 783.4878[M − Glc]^−^; 621.4350[M − (Glc/Glc)]^−^; 161.0435[Glc − H]^−^	Ginsenoside Rd	CG, MCG_12_, MCG_20_	s
83	11.31	C_55_H_92_O_23_	1120.6049	1120.6029	1.7	1143.5941[M + Na]^+^; 831.4874[M − Glc(mal)]^−^	Ginsenoside Rs_1_	CG, MCG_12_, ^##,^* MCG_20_	s
84	11.36	C_42_H_70_O_12_	766.4875	766.4867	1.0	767.4947[M + H]^+^; 605.4423[M − Rha]^+^; 587.4300[M − Rha − OH]^+^; 569.4211[M − Rha − 2 × OH]^+^; 443.3866[M − Rha/Glc]^+^; 425.3769[M − Rha/Glc − OH]^+^; 145.0491[Rha − H − H_2_O]^+^	Ginsenoside Rg_6_	CG, MCG_12_, MCG_20_	[44]
85	11.42	C_51_H_84_O_21_	1032.5515	1032.5505	0.9	1131.5442[M − H]^−^; 765.4785[M − Glc(mal) − OH]^−^; 621.4372[M − (Glc/Glc(mal)]^−^	Malonylginsenoside Rd	CG, MCG_12_, MCG_20_	[45]
86	11.53	C_55_H_92_O_23_	1120.6065	1120.6029	3.0	1165.6047[M + HCOO]^−^; 1077.5851[M − Ac]^−^; 1059.5745[M − Ac − OH]^−^	Ginsenoside Rs_2_	CG, MCG_12_, MCG_20_	s
87	11.69	C_42_H_70_O_13_	782.4738	782.4816	−9.7	805.4631[M + Na]^+^; 621.4354[M − Glc]^+^; 311.0902[Glc/Glc − CH_2_OH]^+^	Ginsenoside Rg_10_	CG, MCG_12_, MCG_20_	a
88	11.79	C_48_H_82_O_18_	946.5494	946.5501	−0.7	991.5476[M + HCOO]^−^; 927.5308[M − OH]^−^; 783.4926[M − Glc]^−^; 621.4412[M − (Glc/Glc) ]^−^	Gypenoside XVII	CG, MCG_12_, MCG_20_	s
89	11.81	C_51_H_84_O_21_	1032.5504	1032.5505	−0.1	1031.5431[M − H]^−^; 987.5535[M − CO_2_]^−^; 621.4412[M − (Glc/Glc(mal))]^−^; 179.0546[Glc − H]^−^	Isomer of malonylginsenoside Rd	CG, MCG_12_, MCG_20_	[49]
90	11.88	C_48_H_82_O_18_	946.5476	946.55021	−2.6	969.5368[M + Na]^+^; 605.4394[M − Glc/Glc]^+^; 587.4312[M − Glc/Glc − OH]^+^; 425.3744[M − Glc/Glc − Glc]^+^; 407.3661[M − Glc/Glc − OH − Glc]^+^	Chikusetsusaponin FK_1_	CG, MCG_12_, MCG_20_	[40]
91	12.18	C_47_H_80_O_17_	916.5398	916.5396	0.2	961.5380[M + HCOO]^−^; 783.4870[M − Xyl]^−^; 621.4388[M − (Xyl/glc)]^−^	Notoginsenoside Fe	CG, MCG_12_, MCG_20_	s
92	12.39	C_50_H_84_O_19_	988.5565	988.5607	−4.1	1011.5458[M + Na]^+^; 831.4819[M − Glc]^+^; 425.3763[M − Glc/Glc(ace) − Glc]^+^	Quinquenoside III	CG, MCG_12_, MCG_20_	[51]
93	12.45	C_47_H_80_O_17_	916.5376	916.5396	−2.1	939.5268[M + Na]^+^; 789.4754[M − 2 × OH − CH_6_O_3_]^+^	Vinaginsenoside R_16_	CG, MCG_12_, MCG_20_	[40]
94	12.59	C_47_H_80_O_17_	916.5361	916.5396	−3.7	939.5253[M + Na]^+^; 407.3672[M − Glc − (Glc/Xyl) − OH]^+^	Gypenoside IX	CG, MCG_12_, MCG_20_	[52]
95	12.91	C_50_H_84_O_19_	988.5569	988.5607	−3.8	1011.5461[M + Na]^+^; 789.4784[M − Glc − 2 × OH]^+^	Quinquenoside III isomer	CG, MCG_12_, MCG_20_	[51]
96	13.29	C_52_H_86_O_19_	1014.5753	1014.5763	−1.0	1037.5645[M + Na]^+^; 857.5032[M − C_4_H_8_O_4_ − 2 × OH]^+^; 393.1376[Glc/Glc(ace) − OH]^+^	Quinquenoside I	CG, MCG_12_, MCG_20_	[53]
97	13.34	C_42_H_72_O_13_	784.4984	784.4973	1.4	829.4966[M + HCOO]^−^; 621.4373[M − Glc]^−^; 161.0437[Glc − H − H_2_O]^−^	Ginsenoside F_2_	CG, MCG_12_, MCG_20_	s
98	13.55	C_42_H_72_O_13_	784.4977	784.4973	0.5	807.4869[M + Na]^+^; 605.4402[M − Glc]^+^; 587.4286[M − Glc − OH]^+^; 425.3765[M − Glc/Glc − OH]^+^; 407.3659[M − Glc/Glc − 2 × OH]^+^	20(*R*)-Ginsenoside Rg_3_	CG, MCG_12_, MCG_20_	s
99	13.57	C_17_H_24_O_2_	260.1774	260.1776	−0.8	261.1847[M + H]^+^	Panaxydol	^∆∆,##^ CG, MCG_12_, MCG_20_	[54]
100	13.77	C_42_H_66_O_14_	794.4464	794.4453	1.4	793.4391[M − H]^−^; 731.4375[M − CO_2_ − OH]^−^; 613.3746[M − Glc]^−^	Chikusetsusaponin Iva	CG, MCG_12_, MCG_20_	[51]
101	14.02	C_52_H_86_O_19_	1014.5750	1014.5763	−1.3	1037.5642[M + Na]^+^; 789.4732[M − Glc − C_2_H_4_O_2_]^+^	Isomer of Quinquenoside I	CG, MCG_12_, MCG_20_	[51]
102	14.38	C_17_H_26_O_3_	278.1879	278.1882	−1.1	279.1952[M + H]^+^	Panaxtriol	CG, MCG_12_, MCG_20_	[55]
103	14.46	C_42_H_72_O_13_	784.4970	784.4973	−0.3	829.4966[M + HCOO]^−^; 621.4373[M − Glc]^−^; 407.3672[M − Glc/Glc − 2 × OH]^+^	20(*S*)-Ginsenoside Rg_3_	CG, MCG_12_, MCG_20_	s
104	15.05	C_18_H_34_O_4_	314.2444	314.2457	−3.8	337.2336[M + Na]^+^	Dibutyl sebacate	CG, MCG_12_, MCG_20_	a
105	17.90	C_16_H_22_O_4_	278.1516	278.1518	−0.7	301.1408[M + Na]^+^; 149.0230[M − C_4_H_9_ − C_4_H_9_O]^+^	*n*-Butyl isobutyl phthalate	CG, MCG_12_, MCG_20_	a
106	17.93	C_30_H_52_O_4_	476.3856	476.3866	−2.2	499.3747[M + Na]^+^; 441.3728[M − 2 × OH]^+^; 423.3590[M − 3 × OH]^+^; 317.2049[M − 2 × CH_3_ − C_8_H_15_O]^+^	20(*R*)-Protopanaxatriol	CG, MCG_12_, MCG_20_	[56]
107	17.95	C_16_H_30_O_2_	254.2246	254.2268	8.2	277.2161[M + Na]^+^	Palmitoleic acid	CG, ^∆,^** MCG_12_, MCG_20_	s
108	18.07	C_19_H_18_O_3_	294.1258	294.1256	0.5	317.1150[M + Na]^+^	Tashinone IIA	CG, MCG_12_, MCG_20_	[57]
109	18.08	C_30_H_48_O_4_	472.3546	472.3553	−1.4	495.3438[M + Na]^+^	β-Amyrone	CG, MCG_12_, MCG_20_	[58]
110	18.08	C_6_H_6_O_3_	126.0331	126.0317	9.4	149.0223[M + Na]^+^	Pyrogallol	CG, MCG_12_, MCG_20_	a
111	18.09	C_30_H_48_O_4_	472.3546	472.3553	−1.8	495.3438[M + Na]^+^	24-Hydroxyoleanolic acid	CG, ^∆∆,^** MCG_12_, MCG_20_	[59]
112	18.09	C_24_H_38_O_5_	406.2720	406.2719	0.3	429.2613[M + Na]^+^; 319.1950[M − CH_3_ − C_4_H_7_O]^+^; 261.2213[M − 2 × C_2_H_4_O_2_ − C_2_H_3_]^+^;	Vitetrifolin	CG, MCG_12_, MCG_20_	a
113	20.14	C_31_H_46_O_2_	450.3535	450.3498	8.0	473.3428[M + Na]^+^; 430.2889[M − C_3_H_7_]^+^	Vitamin K_1_	CG, MCG_12_, MCG_20_	[60]
114	20.97	C_18_H_30_O_2_	278.2224	278.2252	−7.9	277.2151[M − H]^−^; 232.2172[M − COOH]^−^	α-Linolenic acid	^∆∆,##^ CG, MCG_12_, MCG_20_	[61]
115	21.18	C_21_H_38_O_4_	354.2758	354.2770	−3.1	377.2650[M + Na]^+^	β-Monolinolein	CG, MCG_12_, MCG_20_	[62]
116	22.11	C_18_H_32_O	264.2452	264.2453	−0.5	265.2525[M + H]^+^; 149.1320[M − CH_2_ − C_6_H_12_O]^+^; 135.1166[M − CH_2_ − C_7_H_13_O]^+^; 121.1008[M − CH_2_ − C_8_H_15_O]^+^; 109.1010[M − C_8_H_15_O − C_2_H_3_]^+^	(Z)-9,17-Octadecadienal	CG, MCG_12_, MCG_20_	[63]
117	22.49	C_18_H_32_O_2_	280.2386	280.2402	−5.9	279.2313[M − H]^−^; 234.2325[M − COOH]^−^	Linoleic acid	^∆∆,##^ CG, MCG_12_, MCG_20_	s
118	23.85	C_14_H_20_O_2_	220.1478	220.1463	5.6	265.1460[M + HCOO]^−^	Thymyl isobutyrate	CG, MCG_12_, MCG_20_	[64]
119	24.25	C_18_H_34_O_2_	282.2541	282.2559	−6.3	281.2468[M − H]^−^; 236.2481[M − COOH]^−^	9-Octadecenoic acid	^∆∆,##^ CG, MCG_12_, MCG_20_	a
120	24.40	C_36_H_62_O_8_	622.4454	622.4445	1.6	623.4527[M + H]^+^; 316.2842[M − OH − Glc − C_8_H_14_]^+^;	Compound K	CG, MCG_12_, MCG_20_	[40]
121	24.89	C_40_H_56_O_4_	600.4219	600.4179	6.7	601.4292[M + H]^+^; 557.4021[M − C_2_H_4_]^+^	Violaxanthin	CG, MCG_12_, MCG_20_	[65]
122	25.31	C_20_H_38_O_2_	310.2862	310.2872	−3.2	311.2935[M + H]^+^; 277.1995[M − C_6_H_13_]^+^	Ethyloleate	CG, MCG_12_, MCG_20_	a
123	25.35	C_40_H_56_O_4_	600.4212	600.4179	5.6	601.4285[M + H]^+^; 497.3800[M − OH − C_4_H_8_O_2_]^+^	Neoxanthine	CG, MCG_12_, MCG_20_	[65]
124	26.38	C_24_H_38_O_4_	390.2758	390.2770	−2.8	413.2653[M + Na]^+^; 301.1406[M − 3 × C_2_H_5_]^+^; 189.0153[M − C_2_H_5_ − C_4_H_9_ − C_8_H_17_]^+^; 167.0327[M − 2 × C_2_H_17_]^+^	Bis(2-ethylhexyl) phthalate	CG, MCG_12_, MCG_20_	a
125	28.01	C_30_H_46_O_5_	486.3334	486.3345	−2.2	509.3226[M + Na]^+^	Quillaic acid	CG, MCG_12_, MCG_20_	s
126	29.04	C_5_H_8_O_2_	100.0512	100.0524	−10.0	123.0404[M + Na]^+^	Pentanedial	CG, MCG_12_, MCG_20_	[66]

^s^ Identified with standard. ^a^ Compared with spectral data obtained from Wiley Subscription Services, Inc. (USA). ^∆^, ^∆∆^: Represented the content either in CG_4–6 years_ group or in MCG_12 years_ group was significantly higher than the other one (^∆^
*p* < 0.05, ^∆∆^
*p* < 0.001) ^#^,^##^: Represented the content either in CG_4–6 years_ group or in MCG_20 years_ group was significantly higher than the other one (^#^
*p* < 0.05, ^##^
*p* < 0.001) *, **: Represented the content either in MCG_12 years_ group or in MCG_20 years_ group was significantly higher than the other one (* *p* < 0.05, ** *p* < 0).

**Table 3 molecules-24-00033-t003:** The summary table with variable identity, VIP and p value.

Groups for Comparison		Marker’ Name	VIP Value	*p* Value
CG_4–6 years_ vs. MCG_12 years_	CG_4–6 years_	α-linolenic acid	1.23	<0.001
		9-octadecenoic acid	2.17	<0.001
		linoleic acid	2.57	<0.001
		panaxydol	1.49	<0.001
	MCG_12 years_	24-hydroxyoleanolic acid	4.13	<0.001
		ginsenoside F3	2.15	<0.001
		palmitoleic acid	1.54	0.037
CG_4–6 years_ vs. MCG_20 years_	MCG_20 years_	ginsenoside Re1	1.60	<0.001
		ginsenoside Re2	1.75	<0.001
		ginsenoside Rs1	1.59	<0.001
		malonylginsenoside Rb2	4.10	<0.001
		ginsenoside Rf	1.83	<0.001
		isomer of malonylginsenoside Rb1	2.30	<0.001
		quinquenoside R1	1.21	<0.001
	CG_4-6 years_	ginsenoside Ro	1.39	0.017
		isomer of ginsenoside Ro	2.31	0.022
		12,13,15-trihydroxy-9-octadecenoic acid	1.25	0.003
		linoleic acid	7.08	<0.001
		9-octadecenoic acid	3.45	<0.001
		α-linolenic acid	1.86	<0.001
		panaxydol	1.12	<0.001
MCG_12 years_ vs. MCG_20 years_	MCG_12 years_	palmitoleic acid	2.07	<0.001
		24-hydroxyoleanolic acid	3.26	<0.001
	MCG_20 years_	ginsenoside Re1	1.16	0.002
		ginsenoside Rs1	1.89	0.024
		malonylginsenoside Rb2	2.76	0.026
		ginsenoside Re2	1.60	<0.001
		isomer of malonylginsenoside-Rb1	3.87	<0.001

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
