# Peer review of "UPLC-QTOF/MS-Based Nontargeted Metabolomic Analysis of Mountain- and Garden-Cultivated Ginseng of Different Ages in Northeast China"

_molecules, 2018, doi:10.3390/molecules24010033_

Reviewer 1 Report

This manuscript is a very good study on various lots of ginseng. The research is based on some new analysis procedures and a lot of work has been involved by the authors.

I found some English mistakes (especially in the abstract) and typing errors at the end of the 2nd page, the beginning of the 3rd and 4th pages, etc.

I liked very much to find the advantages of your method in "Introduction" and a very good description of all ginseng batches in the "Methods and Materials" section and I like the most that I have found comparative data from 3.2 (page 15) as charts and tables - it is easier to read the results.

I suggest you to mention if you used an instrument to grind and sieve the samples.

Author Response

1. I found some English mistakes (especially in the abstract) and typing errors at the end of the 2nd page, the beginning of the 3rd and 4th pages, etc.

Response: The English mistakes and typing errors all over the text, especially at the above mentioned positions, have been corrected. 

2. I suggest you to mention if you used an instrument to grind and sieve the samples.

Response: During the sample preparation, Baijie stainless steel grinder (BJ-800A, Deqing Baijie Electric Apllicance Co. LTD, Zhejiang Province, China) was used to grind the samples. While the powder was then sieved manually with the Chinese National Standard Sieve No. 3 (R40/3 series). And the above description has been added to the manuscript. 

Reviewer 2 Report

Zhu H et al performed UPLC-QTOF/MS-based non-target metabolomics analysis for cultivated ginseng (CG) and mountain-cultivated ginseng (MCG). These authors found 126 compounds: 85 compounds were detected by ESI positive while 41 were by ESI negative. The majority involves ginsenosides, a group of plant-derived natural sterols. In addition to these, this assay detected other compounds such as triterpenoids, flavonoids, organic acids and organic acid esters. Interestingly, each group of CG(4-6years), MCG(12years), and MCG(20years) has distinct compound profile. Due to the efficacy for classification of CG/MCG, these authors discussed the potential of methodology for future application to different parts of ginseng such as root, leaf, and others.

Major comments:

1)     In discussion, the mechanism of biosynthesis for representative markers and its potential role for plant physiology in CG/MCG needs to be described. For example, lipoxygenases catalyze the oxygenation of unsaturated C18 fatty acid to generate oxygenated fatty acids, such as jasmonic acid, that may be involved in plant maturation through hormone-like substance in some cases.

2)     This is a potentially interesting study visualizing the changes in compounds in CG/MCG by PCA analysis. Thus, Fig 6-9 seemed to be well-organized. In contrast, Fig 10 seemed to be too small: this needs to be enlarged.

Author Response

Reviewer #2

1) In discussion, the mechanism of biosynthesis for representative markers and its potential role for plant physiology in CG/MCG needs to be described. For example, lipoxygenases catalyze the oxygenation of unsaturated C18 fatty acid to generate oxygenated fatty acids, such as jasmonic acid, that may be involved in plant maturation through hormone-like substance in some cases.

Response: It is necessary to describe the mechanism of biosynthesis and the potential role for plant physiology. The supplementary content is as follows:

 Both linoleic acid and α-linolenic acid, the main products of the acetate-malonate pathway, are two essential fatty acids necessary for health. Linoleic acid is used in the biosynthesis of arachidonic acid and thus some prostaglandins, leukotrienes, and thromboxane [68, 69]. Panaxydol, one of the C17 polyacetylenic compounds, origined from acetyl-CoA/ malonyl- CoA via fatty acids with crepenynate as the intermediate [70]. It is a potential antitumor agent due to the significant anticancer activity [71].

A proposed biosynthetic pathway of ginsenosides is as follows: with the action of squalene epoxidase, squalene was converted to 2, 3-oxidosqualene. Dammaranes can be synthesized by dammarenediol synthase, and oleananes by β-amyrin synthase [72]. Ginsenosides was found to have both antimicrobial and antifungal properties. And the molecules are naturally bitter-tasting discouraging insects and other animals from consuming the plant. Ginsenosides likely serve as mechanisms for plant defense [73, 74].

Palmitoleic acid is biosynthesized from palmitic acid by the action of the enzyme Stearoyl- CoA desaturase-1, a key enzyme in fatty acid metabolism [75].

Reference

68. Carvalho E. B. T., Melo I. L. P., Mancini-Filho J.. Chemical and physiological aspects of isomers of conjugated fatty acids. Food Sci. Technol., 2010, 30, 295-307.

69. Harwood J L. Recent advances in the biosynthesis of plant fatty acids. Biochim. et. Biophysica. Acta, 1996, 1301, 7-56.

70. Kim H. S., Lim J. M., Kim J. Y., Park S., Sohn J. Panaxydol, a component of Panax ginseng, induces apoptosis in cancer cells through EGFR activation and ER stress and inhibits tumor growth in mouse models. Int. J. Cancer., 2016, 138, 1432-1441.

71. Nihat K., Elena O., Nicholas S., Huber C., Luis M., Bonfill M. Biosynthesis of Panaxynol and Panaxydol in Panax ginseng. Molecules, 2013, 18, 7686-7698.

72. Liang Y., Zhao S. Progress in understanding of ginsenoside biosynthesis. Plant Biol., 2010, 10, 415-421.

73. Kim Y. J., Zhang D., Yang D. C. Biosynthesis and biotechnological production of ginsenosides. Biotechnol. Adv., 2015, 33, 717-735.

74. Leung K. W., Wong A. S. Pharmacology of ginsenosides: a literature review. Chin. Med., 2010, 5, 20-27.

75. Velisek J., Cejpek K. Biosynthesis of food constituents: Lipids. 1. Fatty acids and derived compounds - a review. Czech J. Food Sci., 2006, 24, 193-216.

2) This is a potentially interesting study visualizing the changes in compounds in CG/MCG by PCA analysis. Thus, Fig 6-9 seemed to be well-organized. In contrast, Fig 10 seemed to be too small: this needs to be enlarged.

Response: Fig 10 in the submitted manuscript is indeed bad-organized. So, Green-Black-Red scheme is re-selected as the color scheme. So, the with this gradient style, the larger values and the smaller values were represented significantly. This Figure has been changed in the text. In addition, there was a typo in the No. of the figures and the tables in the original manuscript, so in the revised version text, the No. of the figures and the tables has been corrected. Such as Fig. 10 in the submitted manuscript, the No. of it has been corrected as Fig. 7.

Reviewer 3 Report

Manuscript title: UPLC-QTOF/MS-based nontargeted metabolomic analysis of mountain- and garden-cultivated ginseng of different ages in Northeast China

In this study, the authors conducted a non-targeted metabolomic analysis on different ginseng samples. UPLC-QTOF was used to collect high resolution MS data in both negative and positive ion modes. Over 120 metabolites were tentatively identified, and multivariate analyses including PCA and OPLS-DA were conducted on the MS data matrixes to illustrate potential sample clustering/separation and determine important components with significant variation. The authors made good efforts in sample collection (3 sample groups in 4 geographical locations) and analysis. The identification of over 120 ginseng metabolites is quite impressive. Multivariate analyses revealed potential compound markers which were proposed to be used in differentiating ginseng samples.

Main concerns:

1. English language need to be carefully checked across the manuscript.

2. Abstract. Abbreviations (e.g. CG, MCG) was used without explanation.

3. Since UNIFI is available, why used Masslynx for data collection?

4. The peak detection details (85 in positive and 41 in negative) in discussion should be moved to result. In Table, some peaks showed mass error more than 5 ppm. For instance, peak 17, identified as lucideric acid had 10 ppm. This is surprisingly high considering the instrument condition. Xevo G2-S with lockmass correction should give a pretty decent accuracy. Some of the peak identification need to be revisited, especially for the ones without any fragmentation data or reference standards.

5. What software was used for PCA and OPLS-DA. Based on the figure style, it looks like SIMCA was used. Also, data pretreatment details need to be provided (data normalization, data scaling etc.).

6. Fig. 6. Figure legend needs adjustment.

7. Fig. 7 – Fig. 10. Some axis labels are missing. Fig. 9, figure caption needs correction.

8. One summary table with variable identity, VIP, p value etc. can be added to summarize the results described in section 3.2. Several markers were determined, and according to authors, their contents were “significantly” higher or lower in certain ginseng groups. What data was used for this comparison, the concentration of compounds, or their ion intensity? Some representative charts showing the differential levels of certain compounds in different ginseng groups can be added.

9. Fig. 10. Resolution need to be improved. The figure was not described or discussed at all in the manuscript.

Author Response

Reviewer #3

1. English language need to be carefully checked across the manuscript.

Response: English language have been carefully checked across the manuscript.

2. Abstract. Abbreviations (e.g. CG, MCG) was used without explanation.

Response: In abstact, the explanation of CG and MCG have been added.

3. Since UNIFI is available, why used Masslynx for data collection?

Response: It is true that Unify can be used to data collection. But in order to obtain the maximum efficiency, we chose Masslynx for data collection and UNIFI for data processing. The reasons are as follows:

1) MassLynx™Software could intelligently control the Waters mass spectrometry system, and provide us with the fundamental platform to acquire, analyze, manage, and share mass spectrometry information.

2) Waters UNIFI Scientific Information System is the first software platform to merge LC and high performance MS data into a single solution that encompasses data acquisition, processing, visualization, reporting, and configurable compliance tools within a networked laboratory environment. UNIFI data and other data, including Masslynx data, can be imported into the UNIFI software.

3) It is feasible to only use UNIFI to collect the data and to process the data. But the very high computer configuration is required. In fact, in our laboratory, UNIFI software and MassLynx software was installated on two separate computers. And the computer for UNIFI software is already with high configuration. But when the UNIFI software running for processing data, the computer speed is really slow. If we use UNIFI to collect and to process the data, the computer will get slower and slower with the more and more data collected. So in order to improve work efficiency, Masslynx was used to collect data, and UNIFI was used for data processing.

4. The peak detection details (85 in positive and 41 in negative) in discussion should be moved to result. In Table, some peaks showed mass error more than 5 ppm. For instance, peak 17, identified as lucideric acid had 10 ppm. This is surprisingly high considering the instrument condition. Xevo G2-S with lockmass correction should give a pretty decent accuracy. Some of the peak identification need to be revisited, especially for the ones without any fragmentation data or reference standards.

Response:

Firstly, the peak detection details (85 in positive and 41 in negative) in discussion has been moved to result. Secondly, after revisited, the value of mass error of peak 111 was miscalculated, and the value has been corrected in the text. Thirdly, there are two kinds of unit, ppm and mDa, used to express mass error as we know. It is indeed that some peaks, such as peak 3 (8.1ppm/ 4.4 mDa), 9 (-6.3ppm/-2.3mDa), 17 (10.0ppm/ 4.8mDa), 20 (9.7ppm/4.5mDa), 21 (7.4ppm/ 4.6mDa), 24 (-9.7ppm/-2.4mDa), 28 (-7.7ppm/-3.8 mDa), 29 (-6.9ppm/-3.5mDa), 32 (-9.3ppm/ 4.5mDa), 53 (-8.5ppm/-3.5mDa), 107 (8.2ppm/-2.2 mDa), 110 (9.4ppm/1.4mDa), 113 (8.0ppm/ 3.7mDa), 114 (-7.9ppm/-2.8mDa), 118 (5.6ppm/1.5 mDa), 119 (-6.3ppm/-1.8mDa), 121 (6.7ppm/ 4.0mDa), 123 (5.6ppm/3.3mDa), 126 (-10.0ppm/ -1.2 mDa), showed mass error more than 5 ppm but less than 5mDa. Forthly, in some literatures on similar research[1,2,3,4], it is reported that 10 ppm has been usually used as the mass error value. In addition, during the revisiting, some fragments were found to have been forgotten to fill in the table. In the revised version, the fragments has been added in the table.

In summary, although the values of mass error of some peaks were more than 5 ppm, these conpoments still be kept in the table.

Reference

1. Wu W., Sun L., Zhang Z., Guo Y. Y., Liu S. Y. Profiling and multivariate statistical analysis of Panax ginseng based on ultra-high-performance liquid chromatography coupled with quadrupole -time-of-flight mass spectrometry. J. Pharm. Bio. Anal. 2015, 107, 141-150.

2. Liang Z. T., Chen Y. J., Liang X., Qin M. J., Yi T., Chen H. B., Zhao Z. Z. Localization of ginsenosides in the rhizome and root of Panax ginseng, by laser microdissection and liquid chromatography–quadrupole/time of flight-mass spectrometry. J. Pharm. Biomed. Anal. 2015, 105, 121-133.

3. Xu X. F., Cheng X. L., Lin Q. H., Li S. S., Jia Z., Han T., Lin R. C., Wang D., Wei F., Li X. R. Identification of mountain-cultivated ginseng and cultivated ginseng using UPLC-TOF MSE with a multivariate statistical sample-profiling strategy. J. Ginseng Res. 2016, 40, 344-350.

4. Xu X. F., Xu S. Y., Zhang Y., Zhang H., Liu M. N., Gao Y., Xue X., Xiong H., Lin R. C., Li X. R. Chemical Comparison of Two Drying Methods of Mountain Cultivated Ginseng by UPLC-QTOF- MS/MS and Multivariate Statistical Analysis. Molecules, 2017, 22, 717-727.

5. What software was used for PCA and OPLS-DA. Based on the figure style, it looks like SIMCA was used. Also, data pretreatment details need to be provided (data normalization, data scaling etc.).

Response: MarkerLynx XS V4.1 software could identify mass-retention pairs present in the samples analyzed. It does so by using a combination of spectral deconvolution, peak integration, and sample alignment. MarkerLynx determines the abundance of each marker in terms of area across all samples and then submits the marker abundance matrix to PCA and OPLS-DA. But the figure style displayed in Extended Statistics (XS) Viewer was not satisfactory. So, Simca software was later used to show the results of PCA and OPLS-DA. But in the text, we forgot to mention the Simca software. In the revised manuscript, the Simca software has been added.

By using MarkerLynx software to obtain marker results, the following steps were performed: acquiring data, creating a MarkerLynx processing method, processing the acquired data and viewing results in XS Viewer. And this step has been added in the text. The main parameters in the method set to process the raw data were list in the manuscript. In the Markerlynx XS training, the processing is shown as the following figure. That means the data pretreatment such as data normalization, data scaling etc., were performed automatically by MarkerLynx software with the creating MarkerLynx processing method.

Figure. The data processing by MarkerLynx software

6. Fig. 6. Figure legend needs adjustment.

Response: The figure legend has been adjusted in the text.

7. Fig. 7-Fig. 10. Some axis labels are missing. Fig. 9, figure caption needs correction.

Response: The missing axis labels has been added in these figures. And the figure caption has been corrected. In addition, there was a typo in the No. of the figures and the tables in the original manuscript, so in the revised version text, the No. of the figures and the tables has been corrected. Such as Fig. 10 in the submitted manuscript, the No. of it has been corrected as Fig. 7.

8. One summary table with variable identity, VIP, p value etc. can be added to summarize the results described in section 3.2. Several markers were determined, and according to authors, their contents were “significantly” higher or lower in certain ginseng groups. What data was used for this comparison, the concentration of compounds, or their ion intensity? Some representative charts showing the differential levels of certain compounds in different ginseng groups can be added.

Response: Firstly, one summary table (Table 3) with variable identity, VIP, p value etc. has been added to summarize the results described in section 3.2. Secondly, It was the ion intensity used for the comparison. Thirdly, the heatmaps was used to show the differential levels of certain compounds in different ginseng groups. And the description about the heatmaps has also been added in the text.

9. Fig. 10. Resolution need to be improved. The figure was not described or discussed at all in the manuscript.

Response: This figure is indeed bad-organized. So, Green-Black-Red scheme is re-selected as the color scheme. As a result, the with this gradient style, the larger values and the smaller values were represented significantly. The figure has been changed in the text. And the description of the figure has been added in the text as follows:

The hierarchical clustering heatmaps, intuitively visualizing the differential levels of potential biomarkers concentration in different ginseng groups, were shown in the figure. The larger contents were represented by red squares and smaller values by green squares. 

Round  2

Reviewer 2 Report

Dear the Editor,

These authors answered questions properly. Thus, Reviewer now considers that this revised manuscript may be acceptable for publication.

Reviewer 3 Report

In the revised manuscript, the authors have properly addressed all the previous concerns. Fig. 2 and Fig. 7 seem to be oddly organized in the manuscript file, maybe due to some software compatibility issue? Except for that the manuscript is acceptable for publication.